# Identification of Transcription Factors Regulating SARS-CoV-2 Tropism Factor Expression by Inferring Cell-Type-Specific Transcriptional Regulatory Networks in Human Lungs

**DOI:** 10.3390/v14040837

**Published:** 2022-04-17

**Authors:** Haonan Tong, Hao Chen, Cranos M. Williams

**Affiliations:** 1Electrical and Computer Engineering, North Carolina State University, Raleigh, NC 27695, USA; htong@ncsu.edu; 2Program in Genetics, Department of Plant and Microbial Biology, North Carolina State University, Raleigh, NC 27695, USA; 3College of Forestry, Shandong Agricultural University, Tai’an 271018, China

**Keywords:** SARS-CoV-2, single-cell RNA-seq, differential network analysis

## Abstract

Severe acute respiratory syndrome coronavirus 2 (SARS-CoV-2) is the virus that caused the coronavirus disease 2019 (COVID-19) pandemic. Though previous studies have suggested that SARS-CoV-2 cellular tropism depends on the host-cell-expressed proteins, whether transcriptional regulation controls SARS-CoV-2 tropism factors in human lung cells remains unclear. In this study, we used computational approaches to identify transcription factors (TFs) regulating SARS-CoV-2 tropism for different types of lung cells. We constructed transcriptional regulatory networks (TRNs) controlling SARS-CoV-2 tropism factors for healthy donors and COVID-19 patients using lung single-cell RNA-sequencing (scRNA-seq) data. Through differential network analysis, we found that the altered regulatory role of TFs in the same cell types of healthy and SARS-CoV-2-infected networks may be partially responsible for differential tropism factor expression. In addition, we identified the TFs with high centralities from each cell type and proposed currently available drugs that target these TFs as potential candidates for the treatment of SARS-CoV-2 infection. Altogether, our work provides valuable cell-type-specific TRN models for understanding the transcriptional regulation and gene expression of SARS-CoV-2 tropism factors.

## 1. Introduction

Coronavirus disease 2019 (COVID-19), caused by SARS-CoV-2, has led to a global public health crisis. Since the first case was reported in late December 2019, SARS-CoV-2 has spread to 215 countries, infected more than 437 million humans, and has led to more than six million deaths, primarily among the elderly [1]. COVID-19 has a high mortality, which is largely due to acute respiratory distress syndrome in the lungs [2]. Since SARS-CoV-2 is mainly transmitted via respiratory droplets, the lung is the preliminary target organ for SARS-CoV-2 infection [3]. Pathological investigations, including postmortem biopsies, have confirmed major pulmonary damage in the lungs as the most likely cause of death, and lifelong damage has been seen in the cases examined [4,5,6]. Understanding how SARS-CoV-2 infects human lung cells is important in order to identify effective treatment options for COVID-19.

SARS-CoV-2 infection in cells (SARS-CoV-2 cellular tropism) is caused by the binding of the viral spike (S) proteins to cellular receptors (e.g., angiotensin-converting enzyme 2; ACE2) and S protein priming by host cell proteases (e.g., transmembrane protease serine protease 2; TMPRSS2) [7,8,9]. The co-expression of ACE2 and TMPRSS2 is widely used as a marker for identifying cells that have the potential to be infected by SARS-CoV-2 [7]. Therefore, many studies have revealed a subset of tissues and cell types that are potentially susceptible to SARS-CoV-2 using single-cell RNA-seq (scRNA-seq) and immunocytochemistry for profiling the co-expression of ACE2 and TMPRSS2 across healthy human tissues [10]. However, ACE2 is not the only functional receptor for SARS-CoV-2. The cell-surface receptors neuropilin-1 NRP1 and CD209 also facilitate the entry of SARS-CoV-2 to specific cells [11,12,13,14]. Additionally, recent studies have tested whether other coronavirus (SARS-CoV, MERS-CoV, and HCoV) receptors work as SARS-CoV-2 receptors. The results have shown that MERS receptor DPP4, SARS-CoV receptors CLEC4G/M, and HCoV-229E receptor CD13 cannot be SARS-CoV-2 receptors because of their inability to bind with SARS-CoV-2 [15,16,17]. The SARS-CoV-2 S protein has been found to be cleaved by several host cell proteases, including CTSL [18], FURIN [19,20], TMPRSS4 [21], and TMPRSS11A [22]. Additionally, in response to virus entry, cells usually express interferon-induced transmembrane proteins (IFITMs) that block the fusion capacity of many enveloped RNA viruses [23]. SARS-CoV-2, however, can hijack IFITM1, IFITM2, and IFITM3 to enhance the infection of the viruses [24,25]; thus, IFITMs serve as factors helping the entry of SARS-CoV-2 into cells. In summary, SARS-CoV-2 cellular tropism depends on the host-cell-expressed proteins (cellular tropism factors), which include receptors (ACE2, NRP1, CD209), proteases (TMPRSS2, 4, 11A, CTSL, and FURIN), and IFITMs (IFITM 1, 2, and 3).

There is a lack of experimental data that identify the transcriptional factors (TFs) that regulate the cellular tropism factors in lung cells. The spatial–temporal expression profiles of these SARS-CoV-2 cellular tropism factor proteins are largely determined by their upstream TFs, whose regulatory roles can differ depending on the cell types involved. The construction of transcriptional regulatory networks (TRNs) can identify potential upstream TFs controlling SARS-CoV-2 tropism factors and elucidate how regulatory relationships are established between TFs and targets [26,27]. The susceptibility of each type of lung cell to SARS-CoV-2 infection can differ widely [18,28,29] and the expression of tropism factors in each cell is heterogeneous [30,31,32], suggesting that there is heterogeneity within each lung cell type in terms of regulating the expression of cellular tropism factors. Therefore, it is essential to construct TRNs in a cell-type-specific way in order to identify the potential regulators controlling SARS-CoV-2 tropism factors in different types of lung cells.

Cell type-specific TRNs that are constructed before and after SARS-CoV-2 infection could be used to compare and identify changes in gene regulation between a healthy and a diseased condition using differential network analysis [33,34]. The differential analysis of TRNs can identify topological differences between networks, which can be quantified using network centrality measures [35]. The most commonly used centrality measures for TRN include degree, betweenness, closeness, pagerank, and eigenvalue [36]. Using these measures, the nodes (TFs) with an important role in regulating target genes can be identified in each network [37,38]. The centrality-based analysis of the TFs in the TRNs of COVID-19-infected donors and healthy donors can not only provide insight into the transcriptional regulation alteration of SARS-CoV-2 tropism after virus infection but can also help identify the candidate nodes (TFs) that may serve as targets for existing drugs to inhibit the virus entry into the human lungs.

Here, we applied a network approach to investigate the regulators of SARS-CoV-2 tropism factors using human lung scRNA-seq data from healthy and COVID-19-infected donors. To study the heterogeneity of the SARS-CoV-2 tropism factors between cell types, we identified cell-type-specific tropism-factor-associated gene modules and potential regulators of these modules specific to each cell type. We then refined the connections within cell-type-specific networks using motif information including ChIP-Seq. This allowed for the identification of candidate TFs that directly regulated the SARS-CoV-2 tropism factors within each cell type. This type of network was constructed for each type of lung cell from healthy donors and COVID-19 patients. Centrality-based differential network analyses were performed for the same type of lung cell across healthy donors and COVID-19 patients and were used to identify how SARS-CoV-2 affected the regulatory signatures of specific cell types. Finally, we applied a centrality-based approach to identify the master TFs from each cell-type-specific TRN that regulate tropism factors. The existing drugs targeting these master TFs were identified as candidates that could be repurposed to mitigate SARS-CoV-2 infection.

## 2. Materials and Methods

### 2.1. Acquisition and Procession of Public Datasets

Single-cell RNA sequence (scRNA-seq) data were retrieved from NCBI, EBI, and the European Genome-Phenome Archive (EGA), including bronchoalveolar lavage fluid (BALF) from 16 samples taken from COVID-19 patients and 16 samples taken from healthy donors. For the donors of COVID-19 patients, 9 samples came from Liao et al. [39] (GSE145926: C141, C142, C143, C144, C145, C146, C148, C149, C152), 2 samples came from Chua et al. [40] (EGAS00001004481: BIHCoV01_BL_1 and BIHCoV04_BL_1), and 5 samples came from Grant et al. [41] (GSE155249: GSM4698176, GSM4698177, GSM4698178, GSM4698180, GSM4698182). For healthy donors, 3 samples came from Liao et al. [39] (GSE145926: C51, C52, C100), 1 sample came from Morse et al. [42] (GSE128033), 4 human bronchial epithelial cell samples came from Lukassen et al. [43] (EGAD00001006185), and 8 samples came from Reyfman et al. [44] (GSE122960: GSM3489182, GSM3489185, GSM3489187, GSM3489189, GSM3489191, GSM3489193, GSM3489195, GSM3489197). The clinical characteristics of the scRNA-seq samples are summarized in Appendix A. All raw data and processed data used for analysis are available upon request.

### 2.2. Assembly of Multiple Distinct scRNA-Seq Datasets

The Seurat framework [45] was used for the comprehensive integration of single-cell data. Quality control was performed for each gene-cell-barcode matrix by examining the mitochondrial contamination (mitochondrial gene percentage < 0.1), the total number of detected molecules (unique molecular identifier count > 1000), and the number of unique genes in a cell (filtering cells that had unique feature counts of less than 200). Feature expression measurements were normalized for each cell by dividing the total counts and multiplying by a scale factor of 10,000, followed by log-transformation (natural logarithm of the given value plus one). For each dataset, the top 2000 variable genes with the highest standardized variance were determined using variance stabilizing transformation [46]. These features were then ranked by the number of datasets they were deemed to be variable in, breaking ties using the median variable feature rank across datasets. The top 2000 genes were selected. The cell pairs were identified by searching for mutual nearest neighbors (MNNs) [47] in a low-dimensional representation (90 dimensions to use) of both datasets from canonical correlation analysis (CCA) [48]. The identified anchors were used to compute all pairwise distances between datasets (total number of cells in the smaller dataset divided by the total number of anchors between the two datasets). The resulting distance matrix was clustered to determine a guide tree. The datasets were then iteratively merged by calculating a transformation matrix based on corresponding anchors [45] for dimensionality reduction and clustering.

### 2.3. Unsupervised Dimensionality Reduction and Clustering

We performed unsupervised dimensionality reduction and clustering according to the method of [45]. A linear transformation was applied by shifting and scaling the expression of each gene to be zero mean and unit variance across cells. Then, a principal component analysis (PCA) was performed on the scaled integrated data. The detected principal components (PCs) were ranked based on their percentage of variance and it was observed that 50 PCs captured the majority of the true signal. Among density-based clustering algorithms that can be used to partition gene expression into arbitrary-sized clusters [49], we used a graph-based clustering analysis, as its clustering results can reflect the cell types involved with a high accuracy [50]. A K-nearest neighbor (KNN) graph of cells was constructed based on the Euclidean distance in PCA space. A shared nearest neighbor (SNN) graph was constructed by calculating the neighborhood overlap (Jaccard index) between every cell and its 20 nearest neighbors in the KNN graph. The Louvain algorithm was used as the modularity optimization technique for extracting communities from the constructed SNN graph [51,52]. The graph-based clustering results were visualized using Uniform Manifold Approximation and Projection (UMAP) based on the identified top 50 PCs.

### 2.4. Marker Gene Selection and the Classification of Cell Types

Marker genes from each cluster were identified via performing Wilcoxon rank sum test on uncorrected gene expressions, which showed an at least 0.25-fold difference (log-scale) between the two groups of cells. We manually annotated cell clusters with broad cell-type classes using the top 2 cluster marker genes compared to the canonical lung cell-type markers annotated in [53]. Clusters without clear markers for distinguishing types were excluded from further analysis.

### 2.5. Module Generation and Direct TF–Target Regulon Identification

Potential regulators were identified for each cell type as TFs whose expression was detected in 0.5% of cells. GRNBoost2 [54] was then used to predict cell-type-specific weighted regulatory interactions, which were then organized into ranking TF–target regulon lists for each of the cell types. A module (i.e., a set of co-expressed TFs and a predicted target gene) was generated based on the top 50 regulators for a gene from the ranking list. Multiple modules were created for each tropism factor gene and their predicted direct regulators. To prune indirect regulatory relationships, the proposed gene modules’ connections validated by the TF–target binding information (with the levels A: high confidence; B: likely confidence; C: medium confidence; D: low confidence taken from [55] used as the TF–target scoring criteria) were considered as direct regulation and kept for further analysis.

### 2.6. Defining the Regulatory Direction of Each TF–Target Regulon

The regulatory direction of our identified direct TF–target regulon was defined based on TF–target data sources [56,57,58]. We extracted the overexpression effect data of TFs from [56], which included transcriptome analyses of cell lines overexpressing each of the 514 human TFs. The data processing was followed by [56] in order to determine the activation or repression effect of each TF on their downstream genes. We extracted the knock out and knock down effect data of TFs from the KnockTF database [57]. Following the database instructions, differentially expressed genes (DEGs) were identified for each TF knock out/knock down using an FDR < 0.05. For each TF perturbation experiment, the TF activity for its downstream DEGs was assigned as activation if the knock out/down resulted in a decrease in gene expression, and vice versa. For TF–target regulons not covered by the above analyses, the TRRUST v2 reference database [58] was used to query their regulatory direction. The regulatory functions of 9396 regulatory interactions of 800 human transcription factors were downloaded from the TRRUST v2 database to annotate predicted TRNs. If there was any conflict regarding the assignment of the type of regulation, we prioritized the signs as follows: overexpression > knock out > knock down > TRRUST v2. For the regulations that were not annotated from these databases, the Pearson product moment correlation between the expression of the TF and its targets was used to determine the type of transcriptional regulation taking place [59]. TF–target pairs that were positively correlated (Pearson correlation ≤ 0.03) were assigned as activating regulation, whereas TF–target pairs that were negatively correlated (Pearson correlation ≤ −0.03) were assigned as repressing regulation.

### 2.7. Network Organization and Visualization

For each cell type, the TF–target regulons were interconnected into a predicted multilevel gene network that was organized and visualized using Cytoscape 3.8.0.

### 2.8. Differential Expression Analysis

A two-sided Wilcoxon rank sum test was performed to identify the differentially expressed genes of each cell type in COVID-19 patients and healthy donors. Bonferroni correction was performed to adjust the *p*-value based on the total number of genes. Genes were determined to be differentially expressed in a cell type if the adjusted *p*-value was less than 0.05.

### 2.9. Differential Network Analysis

Tools from graph theory were used for pinpointing important genes in cell-type-specific function and drivers linked to the disease [36]. In each cell-type-specific gene network, various measures of centrality—namely, degree, betweenness, closeness, eigenvalues, and pagerank—were used to calculate the importance of the TFs. All these centralities were calculated using the package NetworkX 2.4. Single TFs that simultaneously decreased or increased the five centralities from healthy individuals to COVID-19 patients were identified as potential important TFs whose topological features significantly differed between healthy individuals and COVID-19 patients. A cell-type-specific hub TF was identified with maximum difference for any centrality measurement.

Degree centrality is defined as the number of edges that are connected to a point. Our inferred networks are directed. Therefore, the degree of a gene is the sum of the indegree (number of incoming edges) and outdegree (number of outgoing edges) of the gene. Betweenness centrality is calculated from the number of times each node falls in the shortest path of a network. Genes with a higher betweenness will have more control over the network by acting as bridges in the transcriptional cascades. The closeness centrality of a node is measured as the inverse distance between the node and all others in the network. Genes with a high level of closeness have the shortest distance to other genes and thus are able to spread transcriptional information through the network efficiently. In this study, both betweenness and closeness centrality were normalized to compare the network property between networks with different numbers of genes. The eigenvalue centrality of a node is the corresponding element of the principal eigenvector paralleling the largest eigenvalue of the graph adjacency matrix. Genes with a high eigenvalue are likely to be influential in the gene network, since they are connected to other influential genes. Pagerank, a variation of eigenvalue centrality, measures the importance of a node recursively based on the number of incoming edges (indegree) and the quality (i.e., pagerank) of the nodes linking to the node. Different from eigenvalue centrality, pagerank centrality models the probability that the information will continue flowing in the network, as represented by a damping factor (typically 0.85). Pagerank has been used to prioritize TFs in gene regulatory networks with the underlying assumption that genes with a high pagerank are likely to be transcriptionally regulated by [60].

### 2.10. Hybrid Centrality Measures

Three centrality measures, degree, betweenness, and closeness, were combined into one measure that determined the importance of genes in a network [61]. For each gene in a cell-type-specific TRN, the three centrality values were ranked and the consensus measure was calculated as the mean of the centrality ranks.

### 2.11. Mapping Drug Targets

The drugs and their annotations for their mode of action, gene target, and undergone clinical investigations were imported from The Drug Repurposing Hub [62] from the ConnectivityMap (CMap) database [63]. Information on chemical compounds and their corresponding gene targets was extracted from The Probes & Drugs portal [64].

## 3. Results

### 3.1. Expression Landscape of the SARS-CoV-2 Tropism Factors Are Largely Different in Each Cell Type

Previous studies identified receptors (ACE2, NRP1, CD209), proteases (TMPRSS2, 4, 11A, CTSL, and FURIN), and IFITMs (IFITM 1, 2, and 3) as SARS-CoV-2 tropism factors. To investigate the expression specificity of these tropism factors in various cell types within healthy lungs or within SARS-CoV-2-infected lungs, we retrieved public scRNA-seq data from bronchoalveolar lavage fluid (BALF) samples obtained from 16 healthy donors [39,42,43,44] and 16 COVID-19 patients [39,40,41] (Appendix A). We applied a graph-based clustering analysis of the scRNA-seq data from healthy donors and identified 33 distinct cell clusters. These cell clusters were visualized in two-dimensional spaces using the Uniform Manifold Approximation and Projection (UMAP) manifold learning technique (Figure 1a). UMAP is used to construct the fuzzy simplicial set, which is associated with a low-dimensional data representation from manifold approximation [65] and has been used to visualize cell types based on data from scRNA-seq [66]. We classified these clusters into 11 cell types based on their marker genes [53], including club cells, ciliated cells, alveolar epithelial type 1 cells (AT1), alveolar epithelial type 2 cells (AT2), B cells, T cells, natural killer cells (NK), basophil/mast cells, macrophage/monocyte cells, dendritic cells, and endothelial cells (Figure 1a). For scRNA-seq data from COVID-19 patients, we identified 30 clusters that were classified into 7 cell types, including ciliated cells, alveolar epithelial type 2 cells (AT2), B cells, T cells, natural killer cells (NK), macrophage/monocyte cells, and dendritic cells (Figure 1b). The BALFs of COVID-19 patients have higher proportions of immune cells (macrophage/monocyte cell, T cell, B cell, and NK cell) than those obtained from healthy donors and lower proportions of some epithelial cells (ciliated cells and AT2 cells) (Appendix A). We quantified the expression of each gene in the specific cell types based on their percentage of gene-expressing cells and their average expression (Figure 2a,b; Appendix A).

The expression of experimentally validated SARS-CoV-2 tropism factors (ACE2; NRP1; CD209; TMPRSS2, 4, and 11A; CTSL; FURIN; IFITM 1, 2, and 3) was profiled in specific cell types from healthy lungs. The highest proportion of ACE2-expressing cell types was epithelial cell types (club cells, ciliated cells, AT1, and AT2), and most of the CD209-expressed cells were clustered into immune cells (macrophage/monocyte and dendritic cells) (Figure 2a; Appendix A). In contrast with ACE2 and CD209, all healthy lung cell types had a proportion of cells expressing NRP1 (Figure 2a; Appendix A). Our analysis further examined the co-expression of these receptors and their associated proteases in healthy lungs. ACE2 was preferentially co-expressed with TMPRSS2, 4, FURIN, and CTSL in multiple types of epithelial cells (Figure 2; Appendix A) compared to immune cells, consistent with previous studies [21,67,68]. CD209 was mainly co-expressed with CTSL and FURIN in dendritic cells and macrophage/monocyte cells (Figure 2c; Appendix A). Likewise, NRP1 was preferentially co-expressed with CTSL within dendritic cells and macrophage/monocyte cells (Figure 2c; Appendix A). Additionally, we found that IFITM2 and 3 were highly expressed in most lung cell types, while IFITM1 was only preferentially expressed in natural killer cells and T cells (Figure 2a). Next, we analyzed the expression profiles of the tropism factors in the lung cells of COVID-19 patients.

To investigate the effects of virus infection on the expression of tropism factors, the lung cellular expression landscapes of COVID-19 patients were screened on a cell-type-specific level (Figure 2b; Appendix A). With the infection of SARS-CoV-2, most of the ACE2-expressing cells were clustered into AT2 and ciliated cells, and the proportion of cells expressing ACE2 was increased in these two cell types seen in infected lungs compared with in healthy lungs (Figure 2a,b; Appendix A). Likewise, the proportion of CD209 in macrophage/monocyte cells and dendritic cells was increased in the lungs of COVID-19 patients compared to in healthy lungs (Figure 2a,b). By contrast, the proportion of NRP1-expressing cells was decreased in most lung cell types in COVID-19 patients (ciliated cells, B cells, T cells, natural killer cells, macrophages/monocytes, and dendritic cells) compared to the healthy donors (Figure 2a,b; Appendix A). Considering that the entry of SARS-CoV-2 into cells requires the expression of both receptor and proteases, we next investigated the co-expression of receptors (ACE2, CD209, and NRP1) with proteases. Consistent with data from healthy donors, we observed the co-expression of ACE2 with TMPRSS2, 4, FURIN, and CTSL in AT2 and ciliated cells. However, we observed an increased proportion of cells expressing these ACE2–protease pairs in each of the cell types in COVID-19 patients’ lungs (Figure 2d; Appendix A). Compared to the healthy donors, the proportion of macrophage/monocyte cells expressing CD209-CTSL and FURIN was increased, whereas the proportion of dendritic cells expressing CD209-CTSL was unchanged in COVID-19 patients’ lungs (Figure 2d; Appendix A). In contrast, in macrophage/monocyte and dendritic cells, the proportion co-expressing NRP1 with CTSL and FURIN was reduced in COVID-19 patients’ lungs (Figure 2d; Appendix A). Taken together, the differential co-expression pattern of receptor–protease pairs in lung cells between healthy donors and COVID-19 patients suggested an enhanced SARS-CoV-2 tropism in AT2 and ciliated cells via ACE2 and macrophage/monocyte cells via CD209 and a reduced SARS-CoV-2 tropism in macrophages/monocytes and dendritic cells via NRP1 after SARS-CoV-2 infection. For the IFITMs, their expression was increased overall in SARS-CoV-2-affected lung cells compared to cells from healthy lungs (Figure 2a,b, Appendix A). In summary, the expression level of the tropism factors (Figure 2a,b) and the proportion of specific cell types co-expressing each pair of receptors and proteases (Figure 2c,d) differed between the lung cells of healthy donors and those of COVID-19 patients, suggesting that SARS-CoV-2 altered the TRNs regulating tropism factor expression. These results, combined with the differential expression of these tropism factors in each cell type, prompted us to construct TRNs to reveal the regulation of these tropism factors at the cell-type-specific level. We then compared the differential topology of the cell-type-specific TRNs between healthy lung cells and lung cells infected with SARS-CoV-2.

### 3.2. Cell-Type-Specific Trns Affecting Tropism Factors

Three key steps were used for the construction of cell-type-specific TRNs for healthy lung cells and lung cells from COVID-19 patients (Figure 3). In Step 1 (single-cell gene expression analysis), we used the integrated scRNA-seq of lung samples obtained from 16 healthy donors [39,42,43,44] and 16 COVID-19 patients [39,40,41], then used Seurat’s scRNA-seq toolbox to categorize the healthy lung cells into 11 cell types and the COVID-19 patient cells into 7 cell types [45,48]. In Step 2 (TRN inference), we used 1639 human TFs [69] as predefined regulators and 11 experimentally validated SARS-CoV-2 tropism factors as targets for inferring TF–target regulatory networks via the regression tool GRNboost2 [54]. GRNboost2 infers TF–target regulatory networks based on the gene co-expression pattern for each cell type. GRNBoost2 adopts the GENIE3 algorithm, which was the winner of the DREAM5 network challenge, and uses gradient boosting to reduce the time needed to infer a TRN [54]. The direct regulators of the tropism factors are identified by incorporating each target and its 50 top co-expressed TFs as bottom-layered regulators. The network is then expanded by identifying the 50 top co-expressed TFs with bottom-layered regulators as the top-layered regulators. We filter out the indirect TF–target interaction from the candidate regulon based on the ChIP-Seq data and TF-motif binding data [55] (Appendix A). The regulatory direction (activation or repression) of each TF–target is defined based on the public experimental data sources for TF overexpression, TF knock out, and TF knock down, with the data ordered as follows: overexpression > knock out > knock down [56,57,58] (Appendix A). For the TF–target pairs for which there is no experimental evidence, we used in silico correlation analysis to elucidate their regulatory direction. Finally, 18 two-layered TRNs were constructed for the 11 major lung cell types of the healthy donors and the 7 major lung cell types of the COVID-19 patients (Appendix A). In Step 3 (network-based drug repositioning), we quantified the topological features of cell-type-specificTRN across healthy donors and COVID-19 patients via differential network analysis, then used these features to identify the hub TFs that were important for regulating tropism factors in the cells of healthy individuals and COVID-19 patients. The drugs affecting the expression and activity of the identified hub TFs were used as candidates for mitigating the tropism of SARS-CoV-2.

### 3.3. Identification of TFs Directly Regulating Tropism Factors within Each Cell Type

The TRNs from healthy cell types were analyzed and compared to determine how TFs regulate the tropism factors expressed in each healthy cell type. In total, we identified 144 TF–target relationships, including 49 unique TFs directly targeting 11 tropism factors from the networks across all healthy cell types (Appendix A; Figure 4a; Appendix A). We first checked whether the canonical receptor–protease pair ACE2 and TMPRSS2 were directly regulated by a single TF in each specific cell type. Network analyses showed that no TFs had this ability, which suggested that ACE2- and TMPRSS2-mediated SARS-CoV-2 cellular tropism may depend on cooperative regulation by multiple TFs. As such, TMPRSS2 was directly activated by EGR1 and ACE2 was directly activated by c-FOS and repressed by FOXA1 in AT2 cells (Figure 4a). In addition, ACE2 was directly regulated by STAT1 and TMPRSS2 was directly activated by FOXA1 in ciliated cells (Appendix A). Of these four TFs (EGR1, c-Fos, FOXA1, and STAT1), STAT1 and FOXA1 have been shown to regulate ACE2 or TMPRSS2 [70,71,72]. STAT1 is required for SARS-CoV-1 infection [73] and enhances ACE2 expression in mammary tissue during pregnancy [71]. FOXA1 is driven by the androgen receptors and regulates the expression of TMPRSS2 and ACE2 in prostate cancer cells [74,75]. Together with previous studies showing that SARS-CoV-2 virus infection induces EGR1 expression [76] and activates c-FOS via MAPKs using its spike protein [77], we hypothesized that SARS-CoV-2 infection may enhance the tropism of AT2 cells through regulatory cascades of EGR1 and c-Fos and the tropism factors ACE2 and TMPRSS2.

We then asked the question of whether any of the TFs were able to directly regulate any receptor–protease pairs in the individual healthy cell types. Three TFs—CEBPA in monocytes/macrophages (Appendix A), c-Fos in endothelial cells (Appendix A), and STAT1 in monocytes/macrophages (Appendix A) were identified to have this ability. Of endothelial cells co-expressing any receptor–protease pairs, almost 50% of these cells co-express NRP1 and CTSL, which are directly targeted by c-Fos (Appendix A). Of monocyte/macrophage cells co-expressing any receptor–protease pairs, about 79% of these co-express NRP1 and CTSL (STAT1 target), and only 4% of these cells co-express NRP1 and TMPRSS2 (CEBPA target). Our results suggested that the regulons c-Fos-NRP1-CTSL and STAT1-NRP1-TMPRSS2 have the potential to regulate tropism in monocyte/macrophage cells (Appendix A) and endothelial cells (Appendix A), respectively. Additionally, NRP1-CTSL was also identified as the major co-expressed receptor–protease pair in AT1 cells (Appendix A), B cells (Appendix A), basophil/mast cells (Appendix A), club cells (Appendix A), dendritic cells (Appendix A), NK cells (Appendix A), and T cells (Appendix A). Instead of regulating the NRP1-CTSL pair using a single TF, our networks showed that multiple TFs coordinate to regulate NRP1 and CTSL in these healthy cell types, suggesting the robustness of these TRNs in cell types controlling the cellular tropism of SARS-CoV-2.

To further understand how each type of lung cell transcriptionally regulates its cellular tropism after SARS-CoV-2 infection, we analyzed and compared the TRNs of different cell types of COVID-19 patients. From the TRNs, we could identify 35 unique TFs which directly regulate 10 tropism factors through 119 TF–target relationships across patient lung cells (Appendix A; Figure 4b; Appendix A). Of these 35 TFs, 10 TFs (SPI1, USF1, NR2F6, KLF6, HNF4G, MAFB, IRF1, NR1H2, ZNF274, and ELF3) were not determined to regulate tropism factors in the TRNs of healthy cell types. We then identified what tropism factors were directly regulated by these 10 TFs in the lung cell types of COVID-19 patients. Our networks showed that SPI regulates NRP1 and FURIN, HNF4G regulates TMPRSS2, ZNF274 regulates TMPRSS4, and the last seven TFs only target NRP1 in either epithelial cells (AT2, ciliated cells) or immune cells (dendritic, monocytes/macrophages, NK cells, and T cells) (Appendix A). These TFs may be responsible for the dynamic NRP1 mRNA expression change in the lungs of severely ill COVID-19 patients [78]. However, none of these 10 COVID-19 patients TRNs’ TFs regulate receptor–protease pairs in the same cell type from COVID-19 patients. We then queried whether the individual TFs could directly regulate receptor–protease pairs in each of these cell-type-specific TRNs of COVID-19 patients. As a result, five TFs were identified with this regulatory role, including HIF1A, which inhibits ACE2 and FURIN in ciliated cells and monocyte/macrophage cells (Appendix A); c-Jun, which regulates NRP1 and CTSL in dendritic and monocyte/macrophage cells (Appendix A) and CD209 and CTSL in T cells (Appendix A); c-Fos, which regulates NRP1 and CTSL in monocyte/macrophage cells (Appendix A); STAT1, which regulates NRP1 and TMPRSS2 in T cells (Appendix A); and TFAP2C, which regulates CD209 and TMPRSS2 in AT2 cells (Figure 4b). It is noteworthy that c-Jun and c-Fos form an AP-1 early response transcription factor that could be activated by the SARS-CoV-1 nucleocapsid protein [79]. Given the similarity of the sequence identity between SARS-CoV-2 and SARS-CoV-1 in nucleocapsid protein [80], SARS-CoV-2 nucleocapsid protein might also activate the AP-1 early response transcription factor, as SARS-CoV-1 nucleocapsid protein does. With the direct activation of AP-1 (c-Jun and c-Fos) in NRP1–CTSL pairs, monocyte/macrophage cells may be more susceptible to SARS-CoV-2 infection in COVID-19 patients compared to healthy donors. To further understand the transcriptional regulation of the SARS-CoV-2 tropism between lung cell types in healthy and infected states, we quantitatively measured the differential role of TFs in healthy-donor TRNs versus COVID-19-patient TRNs.

### 3.4. Centrality Analysis for Predicting TFs That Are Important in Host Cell Tropism

The regulatory importance of each TF (node) in each network can be evaluated through its centrality (degree, betweenness, closeness, pagerank, and eigenvalues). Knowing the centrality differences between the cell-type-specific TRNs of healthy donors and COVID-19 patients would be helpful for screening out the TFs whose regulatory role is mostly influenced by SARS-CoV-2 infection [36,61]. First, we calculated the centralities of each common TF in the TRNs of healthy individuals and COVID-19 patients for each cell type. Next, we subtracted all the centralities of each common TF in the COVID-19 network and healthy network for the same cell types (AT2, B, ciliate, dendritic, macrophage/monocyte, T, and NK); the TFs showing increases or decreases in the five centralities were used for further analysis (Figure 5; Appendix A). We identified 53 TFs with positive rewiring (increased centrality change) which were more central in COVID-19 networks (Appendix A) and suggested that these TFs might play more important roles in regulating the tropism factors after infection with SARS-CoV-2. We identified 48 TFs with decreased centralities which were more important in the networks of healthy lung cells. To test whether our centrality-identified TFs were functionally associated with SARS-CoV-2 infection, we selected the TFs displaying any of the largest centrality differences between the networks of healthy individuals and COVID-19 patients and assessed their characteristics using the existing literature. A total of 16 unique TFs (FOXP1, STAT3, ATF3, STAT1, SMAD3, ETS1, ETV6, BHLHE40, SPI1, NFKB1, IRF1, RUNX1, MEF2C, RARA, FOS, CREB5) were characterized for further analysis (Appendix A). Of these 16 TFs, 9 TFs (FOXP1, STAT1, STAT3, ATF3, SMAD3, NFKB1, RARA, FOS, IRF1) have been identified as potential drug targets for preventing SARS-CoV-2 infection or inhibiting the activity of SARS-CoV-2 [15,72,81,82,83,84,85,86], suggesting that centrality analysis could be a useful tool for identifying TFs from networks regulating tropism factors.

## 4. Discussion

The expression of host tropism factors is used as a molecular indicator for the susceptibility of lung cells to SARS-CoV-2 [87]. Targeting the transcriptional regulation of tropism factors has been proposed as a viable treatment strategy to prevent SARS-CoV-2 infection [75]. The single-cell sequence of each lung cell type represents a distinct transcriptional landscape [53]. Therefore, it is necessary to elucidate the transcriptional regulatory roadmap in each lung cell type, thereby identifying the TFs regulating tropism factors with the potential to be used as therapeutic targets. We performed a computational analysis of multiple existing public single-cell sequencing datasets of healthy and SARS-CoV-2 infected donors’ lung cells and obtained the count matrix depicting the gene abundances for all single cells. Based on these single-cell data, co-expression modules associated with SARS-CoV-2 host cell tropism factors were created for each lung cell type for healthy donors and COVID-19 patients using GRNboost2. Next, the indirect targets were pruned from these modules based on an available collection of experimentally validated TF–target interactions. Lastly, the activity and direction of these regulons were determined based on the effect of TF perturbation on the regulon’s target genes from the public databases. With these three steps, we established the multiplier hierarchical TRNs targeting tropism factors in a lung-cell-specific manner, which could help us to answer the questions underlying the transcriptional regulatory program for host cell tropism.

### 4.1. Case Study 1: Stat1 Has a Different Regulatory Role for Host Cell Tropism Factors in Healthy vs. SARS-CoV-2-Infected Donors

SARS-CoV-2 N protein has been reported to suppress phosphorylation and the nuclear trafficking of STAT1, thus inhibiting the regulatory activity of STAT1 [88]. Based on our cell-type-specific TRNs, we also identified STAT1 as a direct regulator of tropism factors in some same lung cell types (ciliated, dendritic, and macrophage/monocyte cells) across healthy and infected donors. We then checked whether the regulatory effect of STAT1 on tropism factors was partially diminished in same cell type TRN in healthy and infected states using the STAT1 regulon, which was scored as “A: high confidence” or “B: likely confidence”. For healthy donor TRNs, STAT1 directly targets ACE2 in ciliated cells, CTSL in dendritic cells, and NRP1 and CTSL in macrophage/monocyte cells. In contrast, the direct regulatory activity of STAT1 towards ACE2, NRP2, and CTSL could not be detected in the same cell types’ TRNs from SARS-CoV-2 patients. The results suggest that our TRNs could be used to identify SARS-CoV-2 infection-affected TFs that target tropism factors, providing additional insights for subsequent experiments that extend our knowledge of the virus–host interactions of SARS-CoV-2.

### 4.2. Case Study 2: Differential Centrality Analysis Can Be Used to Identify Important TFs Whose Expression Is Not Differentially Changed between Healthy and SARS-CoV-2-Infected Donors

We applied the specific centrality measures for each TF in the TRNs and identified the TFs whose regulatory role in cell tropism became more pronounced after SARS-CoV-2 infection based on the centrality differences of each TF between healthy donors and COVID-19 patients. Traditionally, differential gene expression analysis between different conditions can help to uncover important regulatory genes [89]. Here, we query whether TFs with an increased/decreased network centrality between healthy and COVID-19 cell-type-specific networks have a corresponding increased/decreased transcriptional expression in each cell type from healthy to SARS-CoV-2-infected lungs. To answer this question, we took the 16 TFs displaying the largest centrality differences between the healthy and COVID-19 networks of the same cell types (FOXP1, STAT3, ATF3, STAT1, SMAD3, ETS1, ETV6, BHLHE40, SPI1, NFKB1, IRF1, RUNX1, MEF2C, RARA, FOS, CREB5) for further analysis. We found that the centrality and transcriptional expression of most TFs in each cell type were consistently increased/decreased from healthy to COVID-19 patients (Appendix A), suggesting that the identification of relative changes in the centrality of single-cell regulatory networks can be used as an alternative to differential expression analysis to classify the biological importance of genes. However, the centrality of eight unique TFs (three of five TFs for B, one of four TFs for ciliated, one of four TFs for macrophage/monocyte, three of four TFs for NK, and one of four TFs for T cells) was not changed in the same direction as expression (centrality decreased with gene expression not decreased; centrality increased with gene expression not increased) (Appendix A). We hypothesized that this inconsistent change between the centrality and expression of these TFs may be attributed to the effect of SARS-CoV-2 virus infection on these TFs at other regulatory levels, such as protein modification. A literature-curated analysis was undertaken to test whether these eight TFs were modified at the protein level. Six TFs (SMAD3, ETS1, RARA, CREB5, FOS, ETV6) could be phosphorylated [90,91,92,93,94,95], and two TFs (BHLHE40 and FOXP1) could be sumoylated [96,97]. It is noteworthy that the phosphorylation of SMAD3 and FOS could be induced by hepatitis virus infection [90,94], as our results supported. In summary, our differential analysis suggested that the transcriptional regulation of SARS-CoV-2 tropism differs between the lungs of healthy individuals and COVID-19 patients within the same cell types. Our results highlight important TFs present within the cells of healthy individuals and COVID-19 patients and further highlight known candidate drugs that target these central TFs.

### 4.3. Case Study 3: Hub Tfs Identified by Network Centrality Analysis Could Be Used as Targets for Drug Repositioning to Prevent SARS-CoV-2 Infection and Transmission

We first ranked each TF within each TRN using its cumulative centrality, which is a combination of degree, betweenness, and closeness (Materials and Methods). The TFs with the top cumulative centrality were then selected as the master regulators in each of the TRNs (Appendix A). From 11 healthy cell-type TRNs and 8 COVID-19 cell-type TRNs, we identified 8 unique central TFs, FOXP1, RUNX1, FOS, SMAD3, JUN, NFKB1, PPARG, and STAT3. Only the top TF, FOXP1, was specifically identified in the TRNs of healthy lung cell types, which includes AT1 (Appendix A), AT2 (Figure 6a), club cells (Appendix A), NK cells (Appendix A), and T cells (Appendix A). In contrast, many TFs were identified specifically as the top TFs in COVID-19 patients’ lung cell types, including SMAD3 in ciliated cells (Appendix A), NFKB1 and JUN in dendritic cells (Appendix A), PPARG in macrophage/monocyte cells (Appendix A), and STAT3 in T cells (Appendix A). In addition, we also identified two TFs serving as the top TFs in the lung cell type TRNs from both healthy individuals and COVID-19 patients. The two TFs were RUNX1 in the TRNs of healthy B cells (Appendix A), basophil/mast cells (Appendix A), dendritic cells (Appendix A), macrophage/monocyte cells (Appendix A), NK cells (Appendix A) and COVID-19 patients’ AT2 cells (Figure 6b) and B cells (Figure 3b) and FOS in the TRNs of healthy ciliated (Appendix A) and endothelial cells (Appendix A) and COVID-19 patients’ ciliated cells (Appendix A) and NK cells (Appendix A). Only FOS in ciliated cells (Appendix A) and RUNX1 in B cells (Appendix A) remained as the top TFs in the TRNs of the same cell type before and after SARS-CoV-2 infection.

We then screened candidate drugs targeting these eight top TFs based on the Drug Repurposing Hub [62]. A total of 67 drugs in clinical and launch stages (Appendix A) could be identified for use in 6 of these 8 TFs (FOS, SMAD3, JUN, NFKB1, PPARG, and STAT3), and no drugs were identified for FOXP1 and RUNX1. We then queried the Probes & Drugs portal [64] to identify the compounds targeting FOXP1 and RUNX1. No compound was identified for FOXP1, and eight compounds targeting RUNX1 were identified (Appendix A). RUNX1 was the only gene targeted by CBFβ Inhibitor. CBFβ Inhibitor has been shown to alter the ability of RUNX1 to bind to target genes and alter their expression, thereby inhibiting the growth of leukemia and breast cancer cell lines [98]. Given that RUNX1 is the top TF for the TRNs of many cell-types, including AT2 cells, the repurposing of the CBFβ Inhibitor to affect RUNX1 may be beneficial for preventing the SARS-CoV-2 infection and transmission. In addition to AT2, ciliated cells are also the primary cells infected by SARS-CoV-2 [99,100]. FOS, as the top TF from healthy ciliated cell TRN, is targeted by the launched drug ephedrine (Appendix A). Ephedrine and its functional analog pseudoephedrine have been used for treating patients with severe COVID-19 pneumonia [101,102]. However, except for FOS, ephedrine regulates 19 other genes; thus, more extensive studies are needed to elucidate the role of FOS and the effect of ephedrine on the treatment of COVID-19 patients. Regarding the TRN of COVID-19 patients’ ciliated cells, its top TF SMAD3 was only identified to be targeted by SIS3. SIS3 inhibits the activity of SMAD3, and it has been shown to have a protective effect on SARS-CoV-2-induced death [103]. We then investigated whether the rest of these 67 drugs specifically target one of the TFs without having any effect on the other genes. We identified T-5224 specifically targeting c-Jun, 15 drugs specifically targeting PPARG, and 8 drugs specifically targeting STAT3 (Appendix A). Additionally, we found that bardoxolone-methyl could affect PPARG and STAT3 together without affecting other genes. Given that Jun, PPARG, and STAT3 are the top TFs from COVID-19 patients’ TRNs, these identified drugs may be candidates for reducing SARS-CoV-2 infection for these cell types.

### 4.4. Limitations of the Present Approach

Firstly, the lung cell types are not completely the same in healthy and SARS-CoV-2-infected donors. Given that SARS-CoV-2 infection can result in the death of lung cells, particularly for several epithelial cell types, such as AT1 cells, club cells, and endothelial cells [104], we could not construct TRNs for all lung cell types and compare the TRNs between healthy and infected donors in our analyses. Secondly, our network construction stochastic algorithm could not guarantee achieving exactly the same results for independent runs on the same input data due to the stochastic nature of the inference approach. The inference approach, the GRNBoost2 algorithm (Step 2 of the Results Section: “Cell-type-specific TRN affecting tropism factors”), is a tree-based stochastic algorithm that uses repeated random samplings of genes to model the expression of target genes for the sets of TFs [105]. Therefore, each regulon composed of TF and target genes might slightly differ between each independent run. We generally expected to achieve consistent results from run to run by averaging the predictions of single regression trees using the gradient boosting machine (GBM) [106]. Thirdly, we employed the information of the TF–target interactions for 1541 human TFs to annotate the direct regulation and pruning of false positives in our predicted single-cell TRNs [55]. This was achieved by sacrificing the predictive power of the TRN models if SARS-CoV-2 could induce a TF–target interaction which was not included in these known TF–target interactions, in which case these regulons would be classified as false negatives in the TRN construction. Fourthly, our pipeline could not infer interactions affected by post-transcriptional regulation. Our GRN boost 2 network inference was based on the principle that if a TF regulates target genes, a change in TF expression will affect the expression of its target genes. However, if the regulatory activity of one TF is not controlled based on its expression but relies on post-translational modifications, such as dimerization with another TF or protein modifications such as phosphorylation, our pipeline could not characterize this type of regulon.

## 5. Conclusions

Overall, this study tried to enrich the knowledge of the transcriptional regulation of SARS-CoV-2 tropism in human lungs by constructing the cell-type-specific TRNs of healthy donors and COVID-19 patients. Our studies comprehensively investigated the expression of 11 tropism factors for 11 healthy lung cell types and 7 cell types from COVID-19 patients. We then integrated co-expression analysis and experimentally validated TF–target information to construct the TRNs regulating these 11 tropism factors in each lung cell type. A total of 11 healthy cell type TRNs and 7 COVID patients’ TRNs were established to visualize the transcriptional regulatory cascades used for activating or repressing the expression of tropism factors. Our differential network analysis elucidated the altered regulatory role of TFs in the same cell types in healthy and SARS-CoV-2-infected states which are responsible for the expression of differential tropism factors. We screened out the key regulators from each TRN and repurposed the drugs targeting these regulators for the potential treatment of SARS-CoV-2 infection. The pipeline described here can also be used to understand the effects of other virus perturbation on cell regulatory networks.

## Figures and Tables

**Figure 1 viruses-14-00837-f001:**
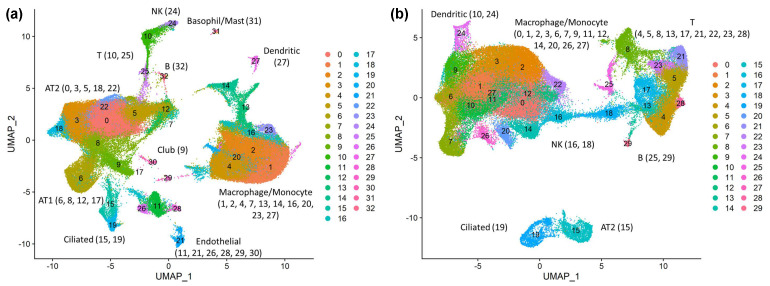
Bronchoalveolar landscapes in healthy donors and COVID-19 patients. (**a**) Uniform Manifold Approximation and Projection (UMAP) presentation of 11 major cell types and associated clusters in BALFs from healthy donors. Healthy cell clusters are annotated with a numeric number and assigned to a cell type. These cell types include club cells, ciliated cells, alveolar epithelial type 1 cells (AT1), alveolar epithelial type 2 cells (AT2), B cells, T cells, natural killer cells (NK), basophil/mast cells, macrophages/monocytes, dendritic cells, and endothelial cells. (**b**) UMAP presentation of 7 major cell types and their associated clusters in BALFs from COVID-19 patients. Infected cell clusters are annotated with a numeric number and assigned to a cell type. Identified cell types include ciliated cells, alveolar epithelial type 2 cells (AT2), B cells, T cells, natural killer cells (NK), macrophage/monocyte cells, and dendritic cells.

**Figure 2 viruses-14-00837-f002:**
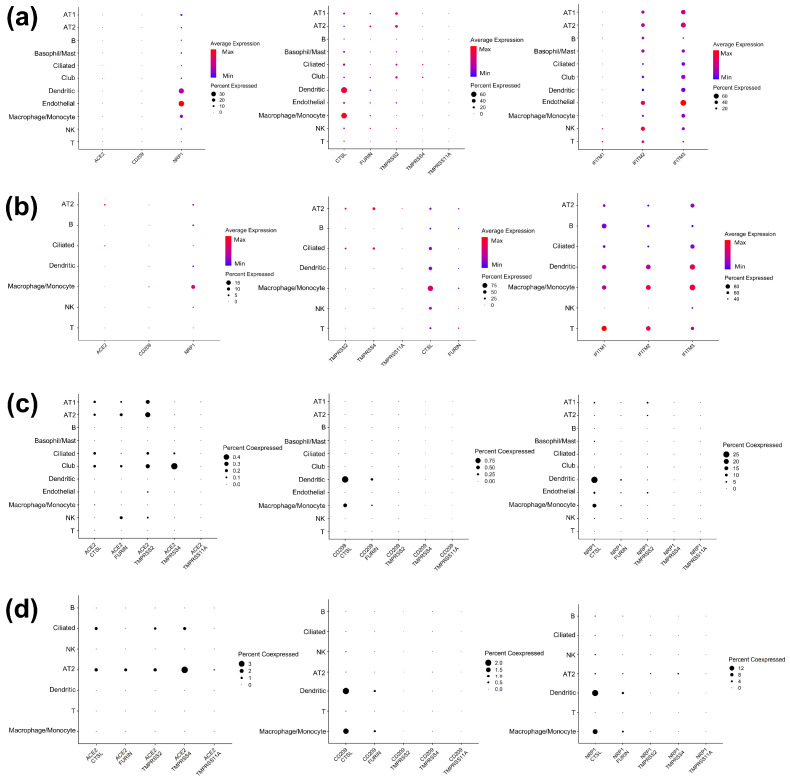
Dot plots showing the expression of SARS-CoV-2 tropism factors in 11 cell types from the BALFs of healthy donors (**a**) and 7 cell types from the BALFs of COVID-19 patients (**b**). These factors include entry receptors (ACE2, CD209, and NRP1; left), entry proteases (CTSL, FURIN, TMPRSS2, 4, and 11A; middle), and interferons (IFITM1, 2, and 3; right). The values of raw expressions were normalized and log-transformed. The size of the dots represents the proportion of cells in each cell type with the expression of each gene, while the color indicates the mean expression of each gene. Dot plots illustrate the co-expression of each receptor–protease pair in 11 cell types from the BALFs of healthy donors (**c**) and 7 cell types from the BALFs of COVID-19 patients (**d**). These receptor–protease pairs include ACE2-CTSL, FURIN, TMPRSS2, 4, and 11A (left); CD209-CTSL, FURIN, TMPRSS2, 4, and 11A (middle); and NRP1-CTSL, FURIN, TMPRSS2, 4, and 11A (right). The size of the dots represents the proportion of cells in each type of co-expressed gene of each receptor–protease pair.

**Figure 3 viruses-14-00837-f003:**
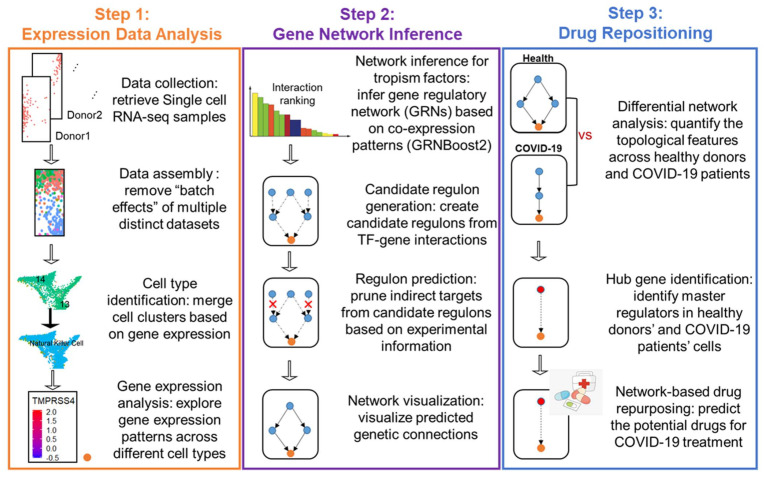
Methodology flowchart. In Step 1, the scRNA-seq of the BALF samples obtained from healthy donors and COVID-19 patients was integrated to identify each cell’s identity and analyze the gene expression within each cell type. In Step 2, the cell-type-specific gene expression was used to reconstruct the transcriptional regulatory network. In Step 3, the roles of TFs were evaluated in the TRNs of each cell type across healthy individuals and SARS-CoV-2 patients, thus providing a clue for repurposing drugs for use in COVID-19 treatment.

**Figure 4 viruses-14-00837-f004:**
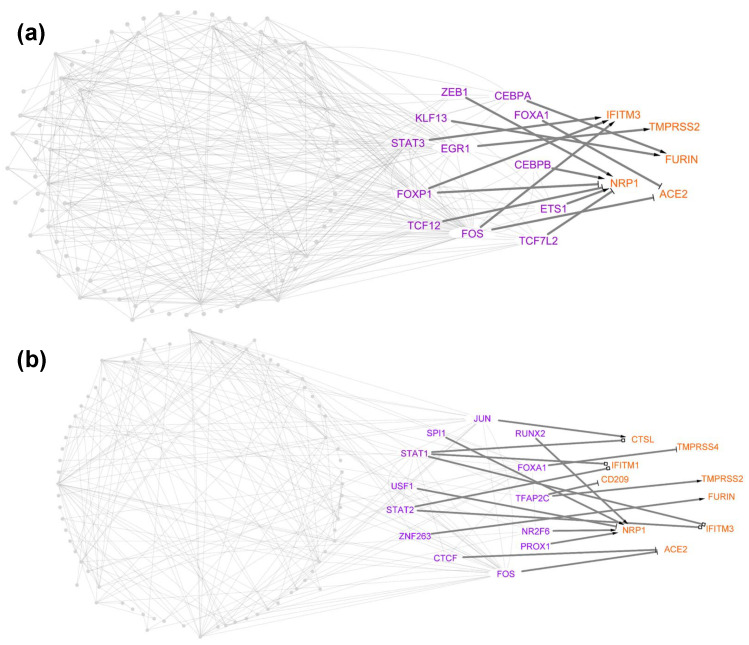
Transcriptional regulatory networks controlling SARS-CoV-2 tropism factors in AT2. Inferred regulation for tropism factors for AT2 in healthy donors (**a**) and in COVID-19 patients (**b**) based on an integrated analysis of in silico and experimental data. The orange nodes represent tropism factors, the purple nodes represent TFs that directly regulate tropism factors, and the gray nodes represent the TFs that indirectly regulate tropism factors. Direct regulations for tropism factors are differentiated between transcriptional activation (delta shape arrows), repression (T shape arrows), and undetermined (square shape arrows).

**Figure 5 viruses-14-00837-f005:**
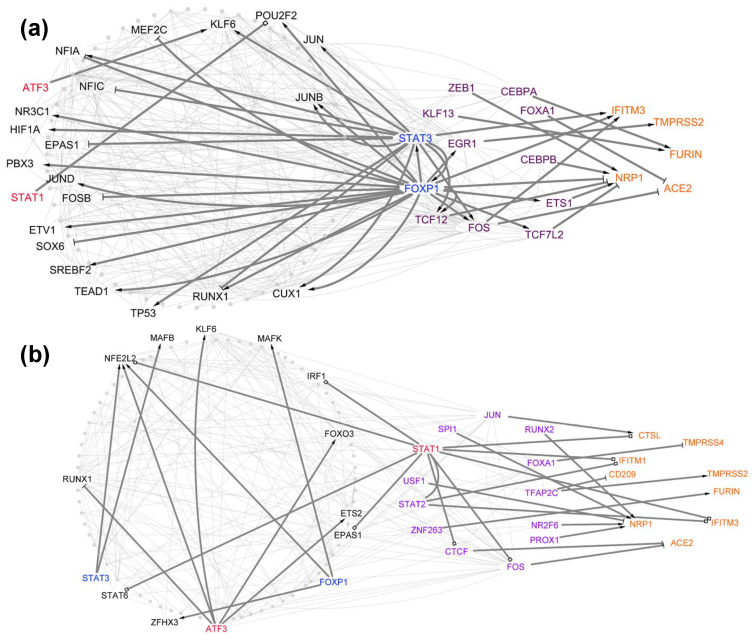
Differential network analysis for ciliated cells. Red (blue) nodes represent TFs that show a significant increase (decrease) in network centralizes from the TRN of healthy donors to the TRN of COVID-19 patients. These red (blue) nodes are marked in the TRN of healthy ciliated cells (**a**) and the TRN of ciliated cells from COVID-19 patients (**b**). The orange nodes represent tropism factors, the purple nodes represent TFs that directly regulate the tropism factors, and the gray nodes represent the TFs that indirectly regulate the tropism factors. Direct regulations of tropism factors are differentiated between transcriptional activation (delta-shaped arrows), repression (T-shaped arrows), and undetermined (square-shaped arrows).

**Figure 6 viruses-14-00837-f006:**
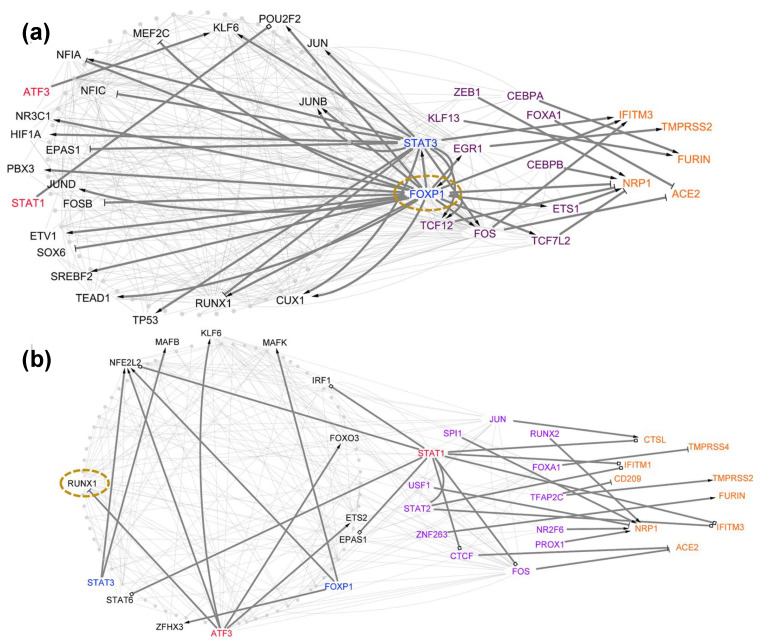
Identification of master TFs in AT2. Dotted circled nodes represent the identified master TFs with the highest hybrid-network centrality. (**a**) FOXP1 is the predicted master TF for healthy donors. (**b**) RUNX1 is the predicted master TF for COVID-19 patients.

## Data Availability

All data were uploaded as Appendix A.

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
