# Peer review of "Identification of Transcription Factors Regulating SARS-CoV-2 Tropism Factor Expression by Inferring Cell-Type-Specific Transcriptional Regulatory Networks in Human Lungs"

_viruses, 2022, doi:10.3390/v14040837_

Round 1
Reviewer 1 Report
The authors substantially improved the manuscript.
Author Response
Reviewer #1 (Comments for the Author and author response):
R#1: The authors substantially improved the manuscript.
We thank the reviewer for their supportive comments.

Reviewer 2 Report
The manuscript titled “Identification of transcription factors regulating SARS-CoV-2 tropism factor expression by inferring cell-type-specific transcriptional regulatory networks in human lungs” by Tong H. et al., is an interesting paper, which describes a computational approach to identify transcription factors regulating the SARS-CoV-2 tropism for different types of lung cells. Transcriptional regulatory networks are also investigated and described in the current paper, using lung single-cell RNA sequencing data.
This topic is of outstanding importance because an in depth understanding of the altered transcriptional regulatory networks during SARS-CoV-2 infection of lung cells, can lead to the identification of potential therapeutic targets. Furthermore, drug repurposing is described in the current paper, suggesting potential candidate molecules for treating SARS-CoV-2 infection.
The paper is well written, figures are informative and well explained. The references are representative of the significant literature concerning the topic of the manuscript.
This is a second revision after resubmission. Previous issues have been fully addressed (description of clinical characteristics of COVID-19 patients and healthy donors; paragraph describing study limitations)
Minor revisions are still required:
-Please be consistent when using acronym for single-cell RNA sequences, considering that both “scRNA-seq” and “scRNAseq” can be found in the manuscript.
-Pearson should be written with capital letter
-Paragraph 4.4 (lines 580-604) should be revised for English language and spelling
Author Response
Reviewer #2 (Comments for the Author and author response):
R#1: Please be consistent when using acronym for single-cell RNA sequences, considering that both “scRNA-seq” and “scRNAseq” can be found in the manuscript.
We thank the reviewer for their supportive comments. We have revised the manuscript and ensured the acronym “scRNA-seq” was used consistently.
R#2: Pearson should be written with capital letter
We thank the reviewer for pointing this out. We have capitalized the first letter of “Pearson” as requested.
R#3: -Paragraph 4.4 (lines 580-604) should be revised for English language and spelling
We thank the reviewer for their supportive comments. The section 4.4 “Limitation of the present approach” has been fully revised to ensure correct English grammar and spelling. Specifically, we utilized a professional English editing service to ensure that the grammar and tone of the entire manuscript is consistent with the standards of the journal.

Reviewer 3 Report
The manuscript has improved, I just strongly suggest careful revision by a mother tongue English speaker or a professional editing service.
Author Response
Reviewer #3 (Comments for the Author and author response):
R: The manuscript has improved, I just strongly suggest careful revision by a mother tongue English speaker or a professional editing service.
We thank the reviewer for their comments. We have made edits to address the reviewer’s concerns. Specifically, we utilized a professional English editing service to ensure that the grammar and tone of the entire manuscript is consistent with the standards of the journal.

This manuscript is a resubmission of an earlier submission. The following is a list of the peer review reports and author responses from that submission.
Round 1
Reviewer 1 Report
The current maunscript simply uses availble data to perform bioinformatics analysis and fails to comply with promises reported in the abstract "we identified the TFs with high centralities from each cell type and repurposed currently available drugs as a potential treatment candidates for SARS-CoV-2 infection". The manuscript lacks functional validation experiments with respect to transcription factors involved (gene knock in/gene knock out) as well as to drug reporposing. Furthermore several spelling mistakes are scattered throughout the manuscript implying low efforts the authors (see for example manuscript title!)
Reviewer 2 Report
The authors performed a computational analysis to identify transcription factors regulating the SARS-CoV-2 tropism for different types of lung cells. Overall the study is well designed and interesting. However, I have only a minor concern regarding the representation of the results.
Comment:
A separate result and discussion section will be better for the readers to understand the study properly. I suggest reducing the results section and adding a new discussion section where the significance of each result should be outlined. Also, the discussion section should highlight what new information this study generated which was not known from similar previous studies.
Reviewer 3 Report
See file attached

Reviewer 4 Report
The manuscript titled “Identification of transcription factors regulating SARS-CoV-2 tropism factor expression by inferring cell-type-specific transcriptional regulatory networks in human lungs” by Tong H. et al., is an interesting paper, which describes a computational approach to identify transcription factors regulating the SARS-CoV-2 tropism for differential types of lung cells. Transcriptional regulatory networks are also investigated and described in the current paper, using lung single-cell RNA sequencing data.
This topic is of outstanding importance because an in depth understanding of the altered transcriptional regulatory networks during SARS-CoV-2 infection of lung cells, can lead to the identification of potential therapeutic target. Furthermore, a currently available drug repurposing is described in the current paper, suggesting potential new treatment candidates for SARS-CoV-2 infection.
The paper is well written, figures are informative and well explained. The references are representative of the significant literature concerning the topic of the manuscript.
Minor revisions are required:
In the title: “transcription” should be changed to “transcription”
Lines 94 and 100: “health donors” should be changed to “healthy donors”
It would be useful to summarize COVID-19 patient and healthy donor clinical characteristics (sex and age, ethnicity, and for COVID-19 patients, pneumonia, ARDS, timing from symptom onset)
A paragraph describing study limitations in the Discussion section is advisable